# Reconstitution of microtubule into GTP-responsive nanocapsules

Noriyuki Uchida[1,2], Ai Kohata [3], Kou Okuro[4,5], Annalisa Cardellini [6], Chiara Lionello [6], Eric A. Zizzi [7], Marco A. Deriu[7], Giovanni M. Pavan [6,8], Michio Tomishige[9], Takaaki Hikima[10] & Takuzo Aida [1,3] ✉

Nanocapsules that collapse in response to guanosine triphosphate (GTP) have the potential as drug carriers for efficiently curing diseases caused by cancer and RNA viruses because GTP is present at high levels in such diseased cells and tissues. However, known GTP-responsive carriers also respond to adenosine triphosphate (ATP), which is abundant in normal cells as well. Here, we report the elaborate reconstitution of microtubule into a nanocapsule that selectively responds to GTP. When the tubulin monomer from microtubule is incubated at 37 °C with a mixture of GTP (17 mol%) and nonhydrolysable GTP* (83 mol%), a tubulin nanosheet forms. Upon addition of photoreactive molecular glue to the resulting dispersion, the nanosheet is transformed into a nanocapsule. Cell death results when a doxorubicin-containing nanocapsule, after photochemically crosslinked for properly stabilizing its shell, is taken up into cancer cells that overexpress GTP.

An ideal nanocarrier for drug delivery would be the one that can selectively collapse to release preloaded drugs in response to endogenous reporters overexpressed in disease tissues[1–9]. Since adenosine triphosphate (ATP) is known to be present at high levels in cancer tissues[10], ATP-responsive nanocarriers might be a promising candidate[3–7]. In 2013, using partially modified biomolecular machine chaperonin GroEL as a monomer, we succeeded in developing a one-dimensional supramolecular polymer that can be depolymerized by the action of ATP to release its cargo[3]. However, ATP is also present in normal cells at rather high concentrations (>1 mM)[11], and thus disease-selective drug delivery using ATP as the endogenous reporter cannot always be ensured. In the present work, we developed a nanocarrier ($^{Cl}$NC$_{GTP/GTP*}$; Fig. 1e) that selectively responds to guanosine triphosphate (GTP). GTP is an intracellular molecule involved in many essential biological processes[12–25], such as cell division[12], nucleotide synthesis[13], and cell signaling[14]. In the cell division process, the tubulin heterodimer (THD), which constitutes microtubules (MTs), uses GTP as an energy source to induce its polymerization and depolymerization[15–20]. GTP is also used as a component for the self-replication of RNA viruses[26–29] such as coronaviruses. Notably, GTP is abundant in certain diseased cells (1.5–4.5 mM)[30] such as rapidly proliferating cancer cells[31] and RNA virus-infected cells[32], whereas the concentration of GTP, unlike that of ATP, is negligibly low in normal cells (<0.3 mM)[33]. Therefore, GTP-responsive nanocarriers have the great potential to efficiently cure cancer and RNA virus-induced diseases including coronavirus disease 2019 (COVID-19)[29]. Although GTP-responsive carriers have already been reported, those carriers also respond to ATP[5]. So far, nanocarriers capable of responding solely to GTP have never been reported.

[1]RIKEN Center for Emergent Matter Science, 2-1 Hirosawa, Wako, Saitama 351-0198, Japan. [2]Department of Applied Chemistry, Graduate School of Engineering, Tokyo University of Agriculture and Technology, 2-24-16 Naka-cho, Koganei, Tokyo 184-8588, Japan. [3]Department of Chemistry and Biotechnology, School of Engineering, The University of Tokyo, 7-3-1 Hongo, Bunkyo-ku, Tokyo 113-8656, Japan. [4]Department of Chemistry, The University of Hong Kong, Pokfulam Road, Hong Kong, China. [5]State Key Laboratory of Synthetic Chemistry, The University of Hong Kong, Pokfulam Road, Hong Kong, China. [6]Department of Applied Science and Technology, Politecnico di Torino, Corso Duca degli Abruzzi 24, 10129 Torino, Italy. [7]PolitoBIOMedLab, Department of Mechanical and Aerospace Engineering, Politecnico di Torino, Corso Duca degli Abruzzi 24, 10129 Torino, Italy. [8]Department of Innovative Technologies, University of Applied Sciences and Arts of Southern Switzerland, Polo Universitario Lugano, Campus Est, Via la Santa 1, 6962 Lugano-Viganello, Switzerland. [9]Department of Physical Sciences, Aoyama Gakuin University, Kanagawa 252-5258, Japan. [10]RIKEN SPring-8 Center, 1-1-1 Kouto, Sayo, Hyogo 679–5198, Japan. ✉e-mail: aida@macro.t.u-tokyo.ac.jp

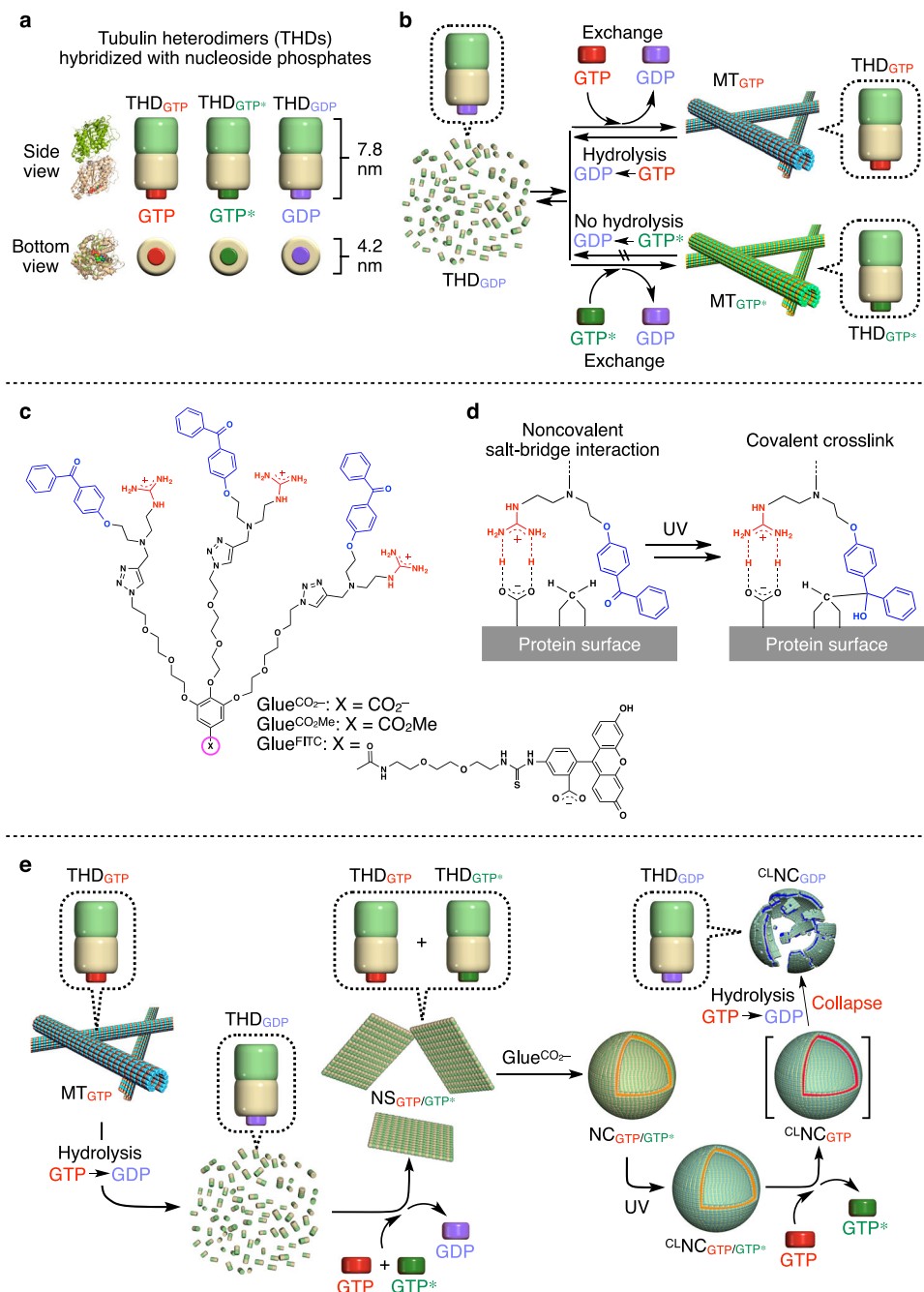

**Fig. 1 | Strategy used to prepare THD-based GTP-responsive $^{CL}NC_{GTP/GTP*}$.** **a** Schematic illustrations of tubulin heterodimers (THDs) hybridized with GTP (THD$_{GTP}$), its nonhydrolyzable analogue GTP* (THD$_{GTP*}$), and GDP (THD$_{GDP}$) at its $\beta$-tubulin unit. **b** Schematic illustration of two self-assembling modes of THD into microtubules (MTs). MT$_{GTP}$ depolymerizes into THD$_{GDP}$ upon GTP hydrolysis. THD$_{GDP}$ rehybridizes with GTP after a GTP treatment, facilitating the formation of MT$_{GTP}$. In contrast, MT$_{GTP*}$ does not undergo depolymerization. **c** Molecular structures of photoreactive molecular glues (Glue$^{CO_2-}$, Glue$^{CO_2-Me}$, and Glue$^{FITC}$) bearing three guanidinium ions (Gu$^+$) and benzophenone (BP) groups at their periphery and $CO_2-$, $CO_2$Me, and FITC groups at the focal core. **d** The molecular glue covalently binds to the protein surface at its photoexcited BP groups after the noncovalent adhesion via a Gu$^+$/oxyanion multivalent salt-bridge interaction. **e** Schematic illustration of the multistep procedure for the synthesis of crosslinked nanocapsules ($^{CL}NC_{GTP/GTP*}$) from MT$_{GTP}$. MT$_{GTP}$ is depolymerized into THD$_{GDP}$, which is incubated with a mixture of GTP* (83 mol%) and GTP (17 mol%) to form nanosheet NS$_{GTP/GTP*}$. Upon treatment with Glue$^{CO_2-}$, NS$_{GTP/GTP*}$ is transformed into spherical nanocapsules (NC$_{GTP/GTP*}$), which are further exposed to UV light, affording $^{CL}NC_{GTP/GTP*}$. Upon addition of GTP, $^{CL}NC_{GTP/GTP*}$ collapses through the conformational change of the THD units induced by GTP hydrolysis.

The nanocapsule (NC) that selectively responds to GTP to release a preloaded drug consists of THD. As shown in Fig. 1a, THD is composed of $\alpha$-tubulin (green) and $\beta$-tubulin (cream), both of which bind to GTP. Notably, GTP attached to the $\alpha$-tubulin unit is neither hydrolysable into GDP nor replaceable with other nucleoside phosphates. In contrast, GTP attached to the $\beta$-tubulin unit is known to be hydrolysable to GDP, which can be replaced with, e.g., GTP*, a nonhydrolysable GTP analogue (guanylyl 5'-$\alpha$,$\beta$-methylenediphosphonate), affording THD$_{GTP*}$ (for convenience, only variable nucleoside phosphates attached to the $\beta$-tubulin unit are shown as a

subscript). Both $THD_{GTP}$ and $THD_{GTP*}$, when heated at 37 °C, have been reported to self-assemble into microtubules $MT_{GTP}$ and $MT_{GTP*}$, respectively (Fig. 1b)[34,35]. Although $MT_{GTP}$ depolymerizes into $THD_{GDP}$ synchronously with the hydrolysis of hybridized GTP to GDP, $MT_{GTP*}$ does not depolymerize into $THD_{GDP*}$ because of the non-hydrolysable nature of GTP*. Therefore, our original motivation was to tackle a challenge of modulating the stability of MTs against depolymerization by changing the $THD_{GTP}$/$THD_{GTP*}$ molar ratio. However, we unexpectedly found that the coassembly of $THD_{GTP}$ and $THD_{GTP*}$ at a certain mixing molar ratio resulted in the formation of a leaf-like 2D nanosheet (NS) rather than MT (Fig. 1e). Because of the increasing importance of 2D objects[36,37], this finding prompted us to functionalize NS using the molecular glue technology[38–40], which we developed for noncovalently functionalizing biomolecules such as proteins, nucleic acids, and phospholipid membranes, and also inorganic materials. Molecular glues are designed to carry multiple guanidinium ion ($Gu^+$) groups and strongly adhere to such biomolecules under physiological conditions by taking advantage of a multivalent salt-bridge interaction with their oxyanionic functionalities (Fig. 1d). For this purpose, we chose $Glue^{CO_2-}$ (Fig. 1c) and incubated it with NS. To our surprise, NS was transformed into a spherical nanocapsule NC (Fig. 1e). Using its photochemically modified version ($^{CL}NC_{GTP/GTP*}$; Fig. 1e), we successfully encapsulated and delivered doxorubicin (DOX)[41], an anticancer drug, into GTP-overexpressing cancer cells to cause cell death.

## Results

### Reconstitution of $MT_{GTP}$ into $NC_{GTP/GTP*}$

Figure 1e illustrates the overall procedure for the synthesis of $NC_{GTP/GTP*}$ from microtubule $MT_{GTP}$. As a typical example of the procedure depicted in the flow chart in Fig. 2a, a 1,4-piperazinediethanesulfonic acid (PIPES) buffer (pH 6.8) solution of $MT_{GTP}$ (5.8 mg $ml^{-1}$, Fig. 2c) was cooled at 4 °C, whereupon $MT_{GTP}$ underwent complete depolymerization within 3 h to yield $THD_{GDP}$ quantitatively (Fig. 2d)[42]. As observed by dynamic light scattering (DLS), the characteristic polydisperse feature of one-dimensional (1D) $MT_{GTP}$ (Fig. 2b, gray) changed to a monodisperse feature with a reduced hydrodynamic diameter of 8 nm (Fig. 2b, blue). Then, $THD_{GDP}$ (0.3 mg $ml^{-1}$) was immersed in a PIPES buffer solution of a mixture of GTP and GTP* (300 μM in total) with a GTP* content of 83 mol% at 37 °C for 30 min. Under the present conditions, $THD_{GDP}$ was converted via the exchange events of GDP → GTP and GDP → GTP* into a mixture of $THD_{GTP}$ and $THD_{GTP*}$, which then spontaneously coassembled into $NS_{GTP/GTP*}$ (Fig. 2e). The small-angle X-ray scattering (SAXS) profile of $NS_{GTP/GTP*}$ showed that its scattering intensity was proportional to $q^{-2}$ in a small $q$ region, which is characteristic of two-dimensional (2D) structures (Supplementary Fig. 9). As determined by atomic force microscopy (AFM), the average thickness of leaf-like $NS_{GTP/GTP*}$ was 5 nm (Fig. 2f). Here, the content of GTP* in the mixture of GTP and GTP* employed for the assembly of $THD_{GDP}$ was critical for its successful transformation into $NS_{GTP/GTP*}$. When the content of GTP* ranged from 85–100 mol%, THD preferentially assembled into MT rather than NS (Supplementary Fig. 10), whereas THD barely assembled when its GTP* content was in the range of 0–70 mol% (Supplementary Fig. 11). Namely, the optimum $THD_{GTP}$/$THD_{GTP*}$ molar ratio for the coassembly into NS is narrow, roughly with a GTP* content of 70–85 mol%. By means of nuclear magnetic resonance (NMR) spectroscopy in DMSO, $NS_{GTP/GTP*}$ prepared at a GTP* content of 83 mol% was found to contain 65 mol% of $THD_{GTP*}$ (Supplementary Fig. 12). It is known that $MT_{GTP}$ and $MT_{GTP*}$, prepared from $THD_{GTP}$ and $THD_{GTP*}$, respectively, are formed by edge-closing of $NS_{GTP}$ and $NS_{GTP*}$ as transient precursors[43]. Note that the longer axis of $THD_{GTP}$ is shorter than that of $THD_{GTP*}$[44]. We suppose that this mismatch possibly affords unfolded $NS_{GTP/GTP*}$ rather than folded $MT_{GTP/GTP*}$. Indeed, when $THD_{GTP*}$ (GTP* content of 83 mol%) was coassembled with $THD_{GTPγS}$ (THD hybridized with guanosine

5′-O-(3-thiotriphosphate), GTPγS), whose length is likewise shorter than $THD_{GTP*}$[44], $NS_{GTPγS/GTP*}$ was formed (Supplementary Fig. 13), whereas the coassembly of $THD_{GTP}$ and $THD_{GTPγS}$ (GTPγS content of 83 mol%), whose longer axes are close in length to each other[44], resulted in $MT_{GTP/GTPγS}$ (Supplementary Fig. 14). $NS_{GTP/GTP*}$ was transformed into $NC_{GTP/GTP*}$ when it was incubated with $Glue^{CO-}$ (100 μM) in PIPES buffer at 37 °C for 30 min (Fig. 1e). This anomalous transformation was accompanied by a large change in the hydrodynamic diameter from 65 nm (Fig. 2b, green) to 660 nm (Fig. 2b, orange) with a slight increase in the zeta potential from −42.2 to −39.0 mV. Transmission electron microscopy (TEM) showed that the newly formed object $NC_{GTP/GTP*}$ was a hollow sphere (Fig. 2g). When $MT_{GTP*}$ and $THD_{GDP}$ instead of $NS_{GTP/GTP*}$ were likewise treated with $Glue^{CO_2-}$, ill-defined agglomerates resulted (Supplementary Figs. 15 and 16).

### Photochemical crosslinking of $NC_{GTP/GTP*}$

The physical stability of $NC_{GTP/GTP*}$ is important for its utilization as a carrier for drug delivery. Through several different experiments, we noticed that $NC_{GTP/GTP*}$ immediately collapsed upon incubation with albumin or serum in buffer, indicating its insufficient stability as a drug carrier. Here, we would like to point out a great advantage of $Glue^{CO_2-}$ and its homologues that their multiple benzophenone (BP) groups upon photoexcitation enable covalent crosslinking with adhering proteins (Fig. 1d). Successful examples so far reported include microtubule and kinesin[39], whose dynamic behaviors could be attenuated by the reaction with photoexcited molecular glues. In the present work, by using fluorescent FITC-appended $Glue^{FITC}$ (Fig. 1c, FITC; fluorescein isothiocyanate) derived from $Glue^{CO_2-}$, we first confirmed that $Glue^{CO_2-}$ has a sufficient photoreactivity with the constituent (THD) of $NC_{GTP/GTP*}$. As shown in Supplementary Fig. 17, the reaction mixture, after being exposed to UV light (300 nm) in PIPES buffer, showed the presence of a fluorescence-emissive covalent adduct between $THD_{GDP}$ and $Glue^{FITC}$ in sodium dodecyl sulfate polyacrylamide gel electrophoresis (SDS-PAGE). Then, we investigated whether this photochemical approach can provide $NC_{GTP/GTP*}$ with a sufficient physical stability by crosslinking the shell. Thus, a PIPES buffer solution of $NC_{GTP/GTP*}$ was exposed to UV light for 2 min, where TEM (Fig. 2h) and AFM imaging results (Supplementary Fig. 18) and DLS profiles (Fig. 2b, red) showed that crosslinked (CL) $^{CL}NC_{GTP/GTP*}$ was spherical and remained intact even upon incubation with albumin (0.1 mg $ml^{-1}$) or serum (0.01%) (Supplementary Figs. 19 and 20). $^{CL}NC_{GTP/GTP*}$, when prepared using $Glue^{FITC}$ instead of $Glue^{CO_2-}$, was fluorescent (Supplementary Fig. 21), indicating the presence of the molecular glue in $^{CL}NC_{GTP/GTP*}$.

### Computational simulation of the assembly of $NS_{GTP/GTP*}$

Considering that tubulin nanosheets $NS_{GTP/GTP*}$ are, on average, 0.04 $μm^2$ wide and 4.2 nm thick, the formation of $NC_{GTP/GTP*}$ (surface area; ~6.2 $μm^2$, membrane thickness; 50 nm) requires at least 1000 pieces of $NS_{GTP/GTP*}$ to assemble. Note that $Glue^{CO_2-}$ carrying both $Gu^+$ and $CO_2-$ groups in its structure can self-assemble via their salt-bridge interaction. In the initial stage of the transformation of $NS_{GTP/GTP*}$ into $NC_{GTP/GTP*}$, we postulate that a certain number of $Glue^{CO_2-}$ molecules utilize their $Gu^+$ groups to form a salt-bridged network with the surface $CO_2-$ groups on $NS_{GTP/GTP*}$ (Fig. 1d) as well as the focal-core $CO_2^-$ group in $Glue^{CO_2-}$. This adhesion event can lower the surface charge density of $NS_{GTP/GTP*}$ and enhance its hydrophobic stacking, which is secured by possible reorganization of the salt-bridged polymeric networks on $NS_{GTP/GTP*}$ (Fig. 1e). We performed all atom molecular dynamics (MD) simulations[45] to explore the adhesion of $Glue^{CO_2-}$ and the effect of this event on the tubulin assembly. From a full MT model (PDB code: 3J6E), we obtained its partial structure composed of three laterally assembled $THD_{GTP*}$ units ($[THD_{GTP*}]_3$) as a model of NS (Fig. 3a). The MD simulation suggested that $Glue^{CO_2-}$ adopts a globular conformation in aqueous media with a hydrodynamic diameter of

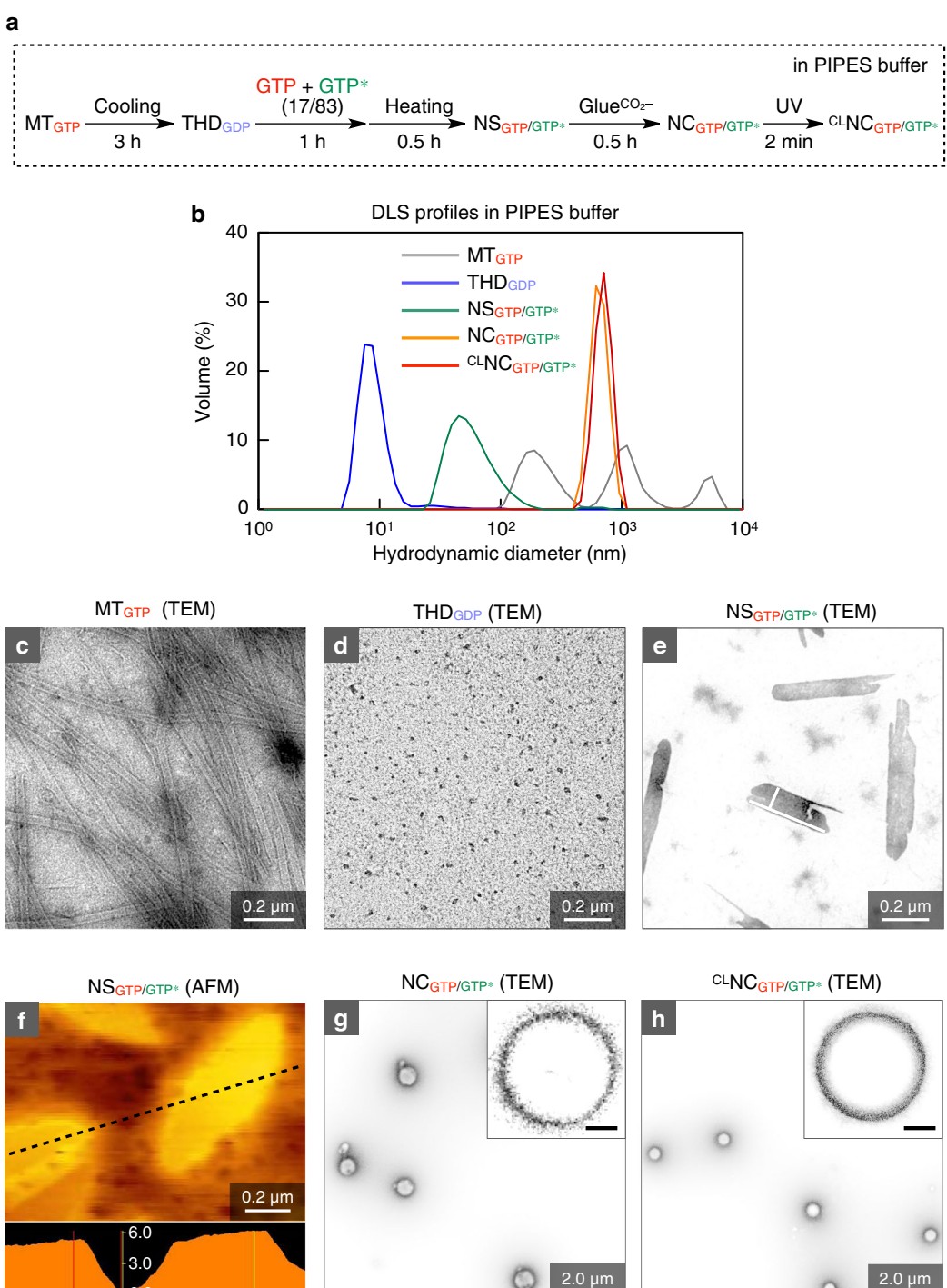

**Fig. 2 | Reconstitution of MT into $^{CL}NC_{GTP/GTP*}$. a** A typical synthetic procedure for the preparation of $^{CL}NC_{GTP/GTP*}$. **b** DLS profiles of $MT_{GTP}$ (gray), $THD_{GDP}$ (blue), $NS_{GTP/GTP*}$ (green), $NC_{GTP/GTP*}$ (orange), and $^{CL}NC_{GTP/GTP*}$ (red) in PIPES buffer. **c–e** TEM images of $MT_{GTP}$ (5.8 mg ml$^{-1}$; **c**), $THD_{GDP}$ (0.3 mg ml$^{-1}$; **d**), and $NS_{GTP/GTP*}$ (0.3 mg ml$^{-1}$; **e**). **f** AFM image of $NS_{GTP/GTP*}$ (0.3 mg ml$^{-1}$) and its height profile. **g, h** TEM images of $NC_{GTP/GTP*}$ (13 µg ml$^{-1}$; **g**) and $^{CL}NC_{GTP/GTP*}$ (13 µg ml$^{-1}$; **h**). All TEM samples were negatively stained with uranyl acetate. Inset scale bars, 250 nm.

1.5 nm (Fig. 3b, Supplementary Fig. 22). When exposed to 30 equivalents of Glue$^{CO_2-}$ (Fig. 3c, d), $[THD_{GTP*}]_3$ enhances its hydrophobic nature (Fig. 3e, f) as a result of the surface charge neutralization by adhering Glue$^{CO_2-}$. In the solvent-accessible surface area of $[THD_{GTP*}]_3$, the hydrophobic dominancy increases from 48% to 57% (Fig. 3g). Notably, when Glue$^{CO_2-}$ was allowed to adhere onto $[THD_{GTP*}]_3$, the molecular simulations suggested that $[THD_{GTP*}]_3$ adopts a slightly more flattened conformation, characterized by a distribution angle with an average value of -156° (Fig. 3h, i, blue), compared with that of

native $[THD_{GTP*}]_3$ (red). The simulations also showed that, even after the Glue$^{CO_2-}$ adhesion, $[THD_{GTP*}]_3$ preserved a certain level of flexibility (Fig. 3i). We also calculated radial distribution functions g(r) between the charged groups of Glue$^{CO_2-}$ and the amino acid residues of $[THD_{GTP*}]_3$. Supposedly, the $CO_2^-$ groups in aspartic acid and glutamic acid are interactive with the Gu$^+$ groups in Glue$^{CO_2-}$, while the cationic groups in lysine and arginine are interactive with the focal $CO_2-$ group in Glue$^{CO_2-}$. As expected, the g(r) data revealed that the Gu$^+$ groups in Glue$^{CO_2-}$ are largely populated near the $CO_2-$ groups on

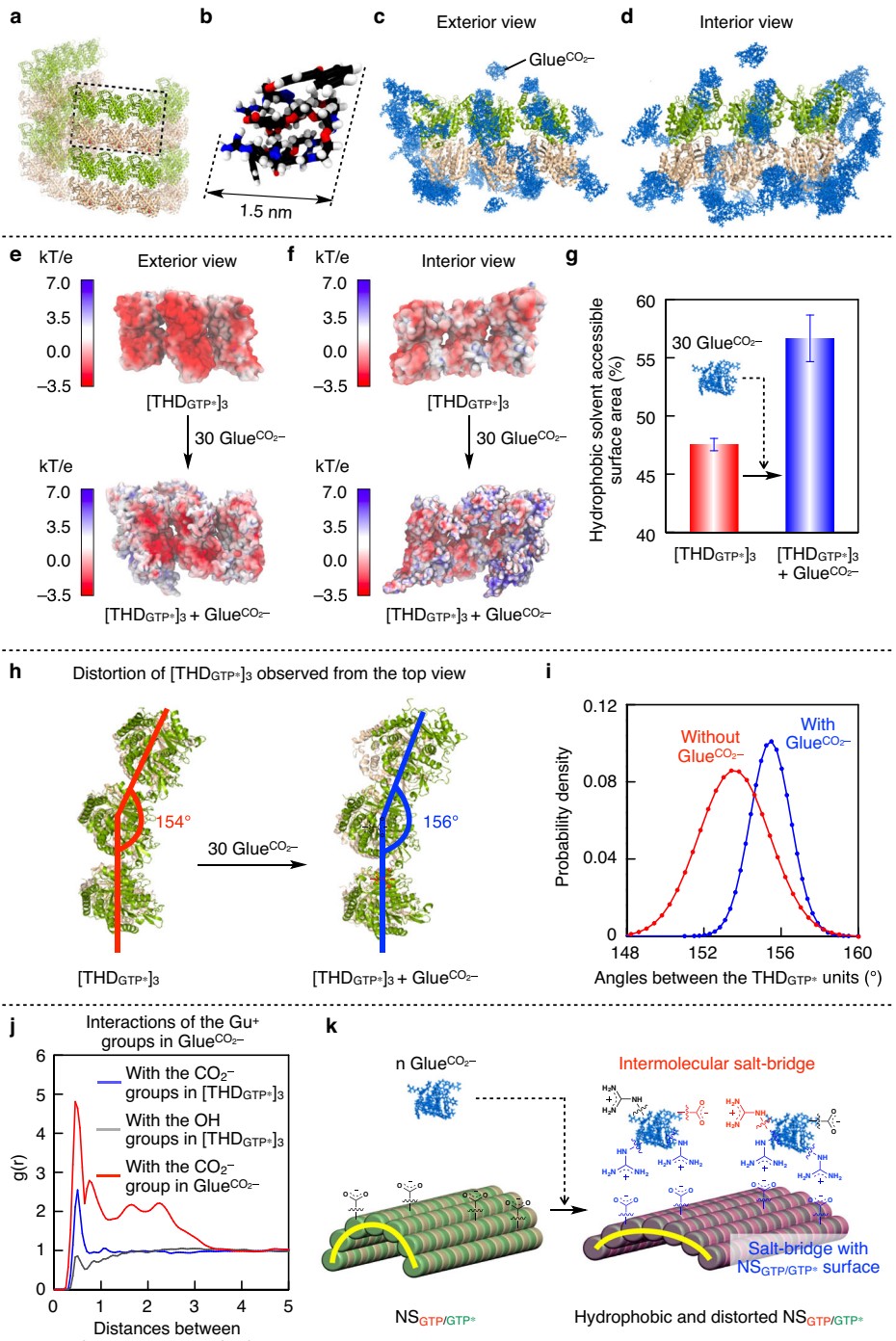

**Fig. 3 | MD simulation of the adhesion events of Glue$^{CO_2-}$ onto the surface of THD$_{GTP*}$.** **a** Three laterally assembled THD$_{GTP*}$ units ([THD$_{GTP*}$]$_3$) in MT$_{GTP*}$ as a partial model of NS. **b** An equilibrated MD snapshot of Glue$^{CO_2-}$. **c, d** The outer (**c**) and inner (**d**) views of [THD$_{GTP*}$]$_3$ hybridized with 30 equivalents of Glue$^{CO_2-}$. **e, f** The outer (**e**) and inner (**f**) views of [THD$_{GTP*}$]$_3$ with its electrostatic surface potential in the absence (upper) and presence (lower) of 30 equivalents of hybridized Glue$^{CO_2-}$. Negative and positive potential areas are colored in red and blue, respectively. **g** The percentage of hydrophobic solvent-accessible surface area in the absence (47.5 ± 0.5; red) and presence (56.7 ± 2.0; blue) of 30 equivalents of hybridized Glue$^{CO_2-}$. Bars represent mean values ± SD from 2000 data points.

**h, i** [THD$_{GTP*}$]$_3$ observed from the top view (**h**) and its angle distributions (**i**) in the absence (red) and presence (blue) of 30 equivalents of hybridized Glue$^{CO_2-}$. **j** Radial distribution functions g(r) of the Gu$^+$ groups in Glue$^{CO_2-}$ with carboxylates (blue) and non-ionic hydroxyl groups (gray) on the [THD$_{GTP*}$]$_3$ surface, and the carboxylate at the focal core of Glue$^{CO_2-}$ (red). **k** Schematic illustration of a possible adhesion event of Glue$^{CO_2-}$ onto NS$_{GTP/GTP*}$ and its effects on the features of NS$_{GTP/GTP*}$. The Gu$^+$ groups in Glue$^{CO_2-}$ form a salt bridge with carboxylates on the NS$_{GTP/GTP*}$ surface and at the focal core of Glue$^{CO_2-}$, and the Glue$^{CO_2-}$-based polymeric network thus formed through this process increases the hydrophobicity of the NS$_{GTP/GTP*}$ surface, making NS$_{GTP/GTP*}$ more flatten.

the $[THD_{GTP*}]_3$ surface (Fig. 3j, blue), whereas they are scarcely populated around the polar but nonionic hydroxyl groups in serine, threonine, and tyrosine (Fig. 3j, gray). Meanwhile, the focal $CO_2^-$ group in $Glue^{CO_2-}$ is not populated around the cationic groups on the $[THD_{GTP*}]_3$ surface (Supplementary Fig. 23). The computational calculation also showed that multiple adhering $Glue^{CO_2-}$ molecules can interact and self-assemble via a salt-bridge interaction between their $Gu^+$ and $CO_2^-$ groups (Fig. 3j, red), which results in forming a dense $Gu^+/CO_2^-$ salt-bridged polymeric network on the $[THD_{GTP*}]_3$ surface (Fig. 3k). This may promote the self-assembly of flexible $NS_{GTP/GTP}$ and stabilize them in the gently curved multilayered configuration of $NC_{GTP/GTP*}$, as observed experimentally[46,47]. As a control experiment, the use of $Glue^{CO_2-Me}$ (Fig. 1c) having a focal ester group instead of its ionized form for the transformation of $NS_{GTP/GTP*}$ into $NC_{GTP/GTP*}$ resulted in an ill-defined agglomerate (Supplementary Fig. 24).

### GTP-responsiveness of $^{CL}NC_{GTP/GTP*}$

We investigated whether photochemically stabilized $^{CL}NC_{GTP/GTP*}$ is responsive to GTP or not. Notably, the concentrations of both extracellular and intracellular GTP are lower than 0.3 mM in normal cells[33]. However, as already described in the introductory part, rapidly proliferating cancer cells and RNA virus-infected cells contain GTP in a concentration range of 1.5–4.5 mM[30]. Therefore, drug-loaded $^{CL}NC_{GTP/GTP*}$, when taken up into such GTP-rich environments, might selectively collapse to release its preloaded guest. Upon incubation for 100 min at 37 °C in PIPES buffer with 0.2 mM GTP, $^{CL}NC_{GTP/GTP*}$ still maintained its spherical shape, as observed by TEM (Fig. 4a). However, when the GTP concentration was increased to 0.5 mM, $^{CL}NC_{GTP/GTP*}$ gradually collapsed (Fig. 4b), displaying a polydisperse DLS profile in 100 min (Fig. 4c, green). This minimum concentration threshold is important for achieving the error-free delivery to GTP-enriched sites. We added Biomol Green™ as a phosphoric acid ($PO_4^-$) detector to a mixture of $^{CL}NC_{GTP/GTP*}$ and GTP (1 mM), and successfully detected $PO_4^-$ by means of electronic absorption spectroscopy, indicating that $^{CL}NC_{GTP/GTP*}$ has a GTPase activity (Fig. 4d). Although $THD_{GTP*}$, the constituent of $^{CL}NC_{GTP/GTP*}$, has no GTPase activity, the product upon incubation of $THD_{GTP*}$ with GTP for 1 h in PIPES buffer at 37 °C showed a GTPase activity comparable to that of $THD_{GTP}$ (Supplementary Fig. 25), indicating the conversion of $THD_{GTP*}$ into $THD_{GTP}$. Thus, under the GTP-rich conditions described above, GTP* in $^{CL}NC_{GTP/GTP*}$ is likely replaced with GTP to afford $^{CL}NC_{GTP}$, which possibly collapses along with the hydrolysis of GTP in a manner analogous to the depolymerization of $MT_{GTP}$. Of particular importance, $^{CL}NC_{GTP/GTP*}$ remained intact to the treatment with other triphosphates (0.5 mM), such as ATP, cytosine triphosphate CTP, uracil triphosphate UTP (Fig. 4e, Supplementary Figs. 26 and 27).

### Guest encapsulation into $^{CL}NC_{GTP/GTP*}$

How to stably encapsulate guests inside nanocarriers is one of the important subjects for drug delivery. By using gold nanoparticles ($NP_{Au}$; 14 pM, diameter 50 nm) as a guest, we succeeded in obtaining $NP_{Au}$-encapsulated $^{CL}NC_{GTP/GTP*}$ by adding $Glue^{CO_2-}$ (100 μM) to a PIPES buffer solution of a mixture of $NS_{GTP/GTP*}$ (13 μg ml$^{-1}$) and $NP_{Au}$ at 37 °C. After 30-min incubation, the resulting mixture was exposed for 2 min to UV light (300 nm) for crosslinking. Using TEM (Fig. 4f) and asymmetric field flow fractionation analysis (Supplementary Fig. 28), we confirmed that $^{CL}NC_{GTP/GTP*}$ encapsulated $NP_{Au}$ ($^{CL}NC_{GTP/GTP*} \supset NP_{Au}$) in its hollow sphere. We also confirmed that $^{CL}NC_{GTP/GTP*}$, when treated with GTP, indeed released its preloaded guest. For this purpose, we first prepared FITC-labeled $THD_{GDP}$ with a mixture of GTP and GTP* (GTP* content: 83 mol%) in PIPES buffer, and further incubated the resulting fluorescent $NS_{GTP/GTP*}$ with $Glue^{CO_2-}$ in the presence of doxorubicin (DOX) for 30 min. Then, the mixture was exposed for 2 min to UV light (300 nm) for transforming $NC_{GTP/GTP*} \supset DOX$ into $^{CL}NC_{GTP/GTP*} \supset DOX$, which was confirmed by confocal laser scanning microscopy (CLSM) to carry both

FITC and DOX dyes (Supplementary Fig. 29, green and red, respectively). When $^{CL}NC_{GTP/GTP*} \supset DOX$ was incubated with 1 mM GTP in PIPES buffer for 100 min, DOX, as observed by CLSM, became much less fluorescent, indicating the disruption of $^{CL}NC_{GTP/GTP*}$ to release DOX (Fig. 4g, (i)–(iii), lower panel). Upon incubation for 20 min, 50 min, and 100 min, the residues obtained by ultrafiltration (cut-off molecular weight = 5000) of the reaction mixtures contained 73%, 53%, and 21% of the total amount of preloaded DOX, respectively (Fig. 4h), while in the absence of GTP, DOX was not released (Fig. 4g, (i)–(iii), upper panel).

### Intracellular drug delivery with $^{CL}NC_{GTP/GTP*}$

As a proof-of-concept study, we investigated whether FITC-labeled $^{CL}NC_{GTP/GTP*}$ can be taken up by human hepatocellular carcinoma Hep3B cells (Fig. 5a). The cells were incubated in Eagle's minimum essential medium (EMEM) containing $^{CL}NC_{GTP/GTP*}$ (0.5 μg ml$^{-1}$) for 2.5 h, rinsed with Dulbecco's phosphate-buffered saline (D-PBS), and further incubated in EMEM containing 10% fetal bovine serum (FBS) for 1.5 h. CLSM (Fig. 5b (i), left panel) together with flow cytometry analysis (Fig. 5c) revealed that most of the cells took up FITC-labeled $^{CL}NC_{GTP/GTP*}$. Upon subsequent incubation for 21.5 h in EMEM (10% FBS), the entire cytoplasm eventually became fluorescent (Fig. 5b (ii), right panel) as a possible consequence of the collapse of incorporated $^{CL}NC_{GTP/GTP*}$. In sharp contrast, FITC-labeled $THD_{GDP}$ and $NS_{GTP/GTP*}$, the intermediates for constructing $^{CL}NC_{GTP/GTP*}$, were scarcely taken up into Hep3B cells (Supplementary Figs. 30 and 31). The high intracellular uptake of FITC-labeled $^{CL}NC_{GTP/GTP*}$ is possibly due to a salt-bridge interaction between the $Gu^+$ groups in adhering $Glue^{CO_2-}$ and cell-surface oxyanionic groups[48]. We confirmed that the intracellular uptake was little affected by the presence of endocytosis inhibitor $NaN_3$ (ref. [49]), suggesting that the incorporation of $^{CL}NC_{GTP/GTP*}$ into Hep3B cells was caused via an endocytosis-independent direct pathway (Supplementary Fig. 32). For the drug delivery application of $^{CL}NC_{GTP/GTP*}$, we conducted a cell viability assay with $^{CL}NC_{GTP/GTP*} \supset DOX$. When treated with $^{CL}NC_{GTP/GTP*} \supset DOX$ ($[^{CL}NC_{GTP/GTP*}]$ = 2.6 μg ml$^{-1}$, [DOX] = 2 μM) in EMEM (Fig. 5d) for 2.5 h, Hep3B cells took up DOX as observed by CLSM after a subsequent incubation in EMEM (10% FBS) for 1.5 h (Fig. 5e (iii), left panel), and then died within next 21.5 h to form an ill-defined agglomerate (Fig. 5e (iv), right panel). We also confirmed that Hep3B cells took up a larger amount of DOX in $^{CL}NC_{GTP/GTP*}$ (Fig. 5f, red) than DOX alone (Fig. 5f, orange). Accordingly, $^{CL}NC_{GTP/GTP*} \supset DOX$ successfully lowered the cell viability to 30 ± 6% (Fig. 5g, red), whereas that caused by DOX alone was only 48 ± 15% (Fig. 5g, orange). As expected, the cell viability decreased as the concentration of $^{CL}NC_{GTP/GTP*} \supset DOX$ was increased (Supplementary Fig. 33), while the viability upon incubation with $^{CL}NC_{GTP/GTP*} \supset DOX$ did not substantially increase when the incubation time was shortened from 2.5 h to 1.0 h (Supplementary Fig. 34). This is likely caused by the GTP-selective collapse of $^{CL}NC_{GTP/GTP*}$. The intracellular delivery of $^{CL}NC_{GTP/GTP*} \supset DOX$ was also successful with other cell lines such as A549 cell and HeLa cell (Supplementary Fig. 35). We also confirmed that neither the coexistence of $THD_{GDP}$ nor $THD_{GDP}/Glue^{CO_2-}$ enhanced the efficacy of DOX (Supplementary Fig. 36). Together with the noncytotoxic nature of $^{CL}NC_{GTP/GTP*}$ (Fig. 5g, green) and its stability in a range of pH at tumor tissue (Fig. 2h)[50], these results allow us to expect that $^{CL}NC_{GTP/GTP*}$ may have the potential to deliver preloaded drugs into cancer cells using GTP as an endogenous reporter.

## Discussion

Here, we have documented the successful reconstitution of $MT_{GTP}$ into a GTP-responsive nanocarrier (Fig. 1). $MT_{GTP}$ is depolymerized into $THD_{GDP}$, which is incubated with a mixture of GTP* and GTP (content of GTP*: 70–85 mol%), thereby facilitating the in situ coassembly of the resulting $THD_{GTP*}$ and $THD_{GTP}$ monomers to form $NS_{GTP/GTP*}$. Subsequently, $NS_{GTP/GTP*}$ is treated with molecular glue $Glue^{CO_2-}$ to be transformed into spherical $NC_{GTP/GTP*}$, followed by UV exposure to afford crosslinked $^{CL}NC_{GTP/GTP*}$ capable of stably

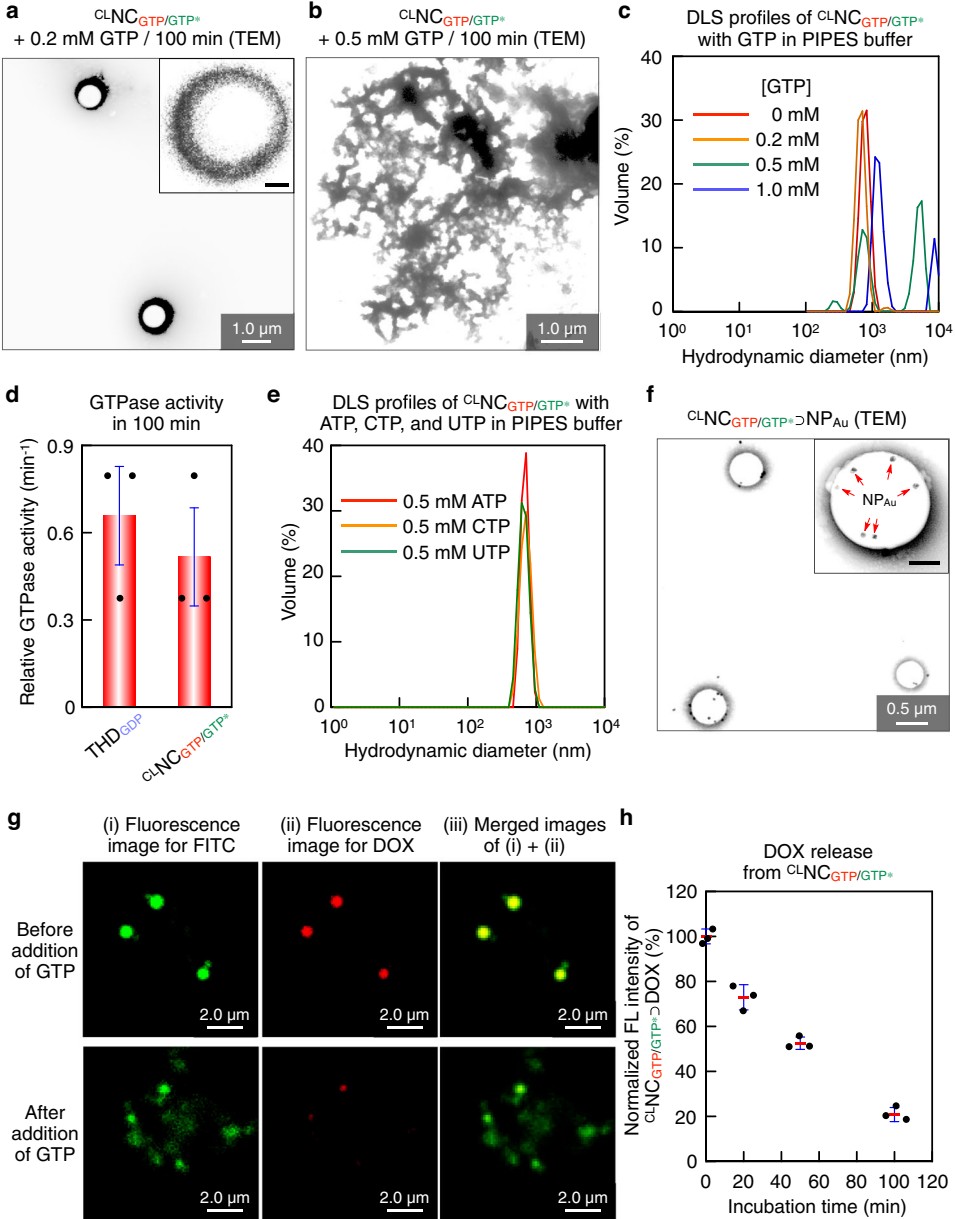

**Fig. 4 | GTP-triggered collapse of $^{CL}NC_{GTP/GTP*}$. a, b** TEM images of $^{CL}NC_{GTP*}$ after a 100-min incubation with GTP at its concentrations of 0.2 mM (**a**) and 0.5 mM (**b**). **c** DLS profiles of $^{CL}NC_{GTP/GTP*}$ (8.7 µg ml$^{-1}$) in PIPES buffer after a 100-min incubation with GTP at its concentrations of 0 mM (red), 0.2 mM (orange), 0.5 mM (green), and 1 mM (blue). **d** GTPase activities of $THD_{GDP}$ (left) and $^{CL}NC_{GTP/GTP*}$ (right) in PIPES buffer. The data was obtained from three biologically independent samples ($n = 3$). **e** DLS profiles of $^{CL}NC_{GTP/GTP*}$ (8.7 µg ml$^{-1}$) in PIPES buffer after a 100-min incubation with 1 mM of ATP (red), CTP (orange), and UTP (green). **f** TEM image of $^{CL}NC_{GTP/GTP*} \supset NP_{Au}$ ([$^{CL}NC_{GTP/GTP*}$] = 13 µg ml$^{-1}$, [$NP_{Au}$] = 13 pM). **g** CLSM images of FITC-labeled $^{CL}NC_{GTP/GTP*} \supset DOX$ ([$^{CL}NC_{GTP/GTP*}$] = 13 µg ml$^{-1}$, [DOX] = 10 µM) incubated without (upper panel) and with (lower panel) 1 mM GTP at 37 °C for 100 min. Micrographs display locations of FITC (i, green) and DOX (ii, red), and their merged images (iii). Scale bars, 2.0 µm. **h** Fluorescence intensities at 590 nm ($\lambda_{ext}$ = 470 nm) of residual DOX obtained after 20, 50, and 100-min incubations of a PIPES solution of $^{CL}NC_{GTP/GTP*} \supset DOX$ with 1 mM GTP, followed by ultrafiltration. Red bars represent mean values ± SD from three different samples.

encapsulating guests (Fig. 2). In GTP-rich environments, $^{CL}NC_{GTP/GTP*}$ collapses and releases preloaded guests through the transformation of $^{CL}NC_{GTP/GTP*}$ into $^{CL}NC_{GTP}$ followed by the hydrolysis of its bound GTP into GDP, analogous to the depolymerization of $MT_{GTP}$ (Fig. 4). Using $^{CL}NC_{GTP/GTP*}$, we successfully delivered DOX into cancer cells that overexpress GTP, and caused cell death more efficiently than DOX alone (Fig. 5). Most importantly, $^{CL}NC_{GTP/GTP*}$ is a drug carrier that can selectively collapse in response to GTP rather than ATP that is abundant in normal cells. Since cells infected with RNA viruses such as coronavirus produce a large amount of GTP in their self-replication process, GTP is an endogenous reporter for RNA virus-infected cells. In vivo utilization of $^{CL}NC_{GTP/GTP*}$ for curing RNA virus-

induced diseases such as COVID-19 is one of the interesting subjects worthy of further investigation.

## Methods
MD simulation was performed using AmberTools 20, GROMACS 2020.5 package, and Visual Molecular Dynamics (VMD) package, and MD simulation methodologies are described in the Supplementary Information.

### Reconstitution of $MT_{GTP}$ into $^{CL}NC_{GTP/GTP*}$
$THD_{GTP}$ was obtained by purification from porcine brain[51] by two cycles of polymerization and depolymerization in PIPES buffer

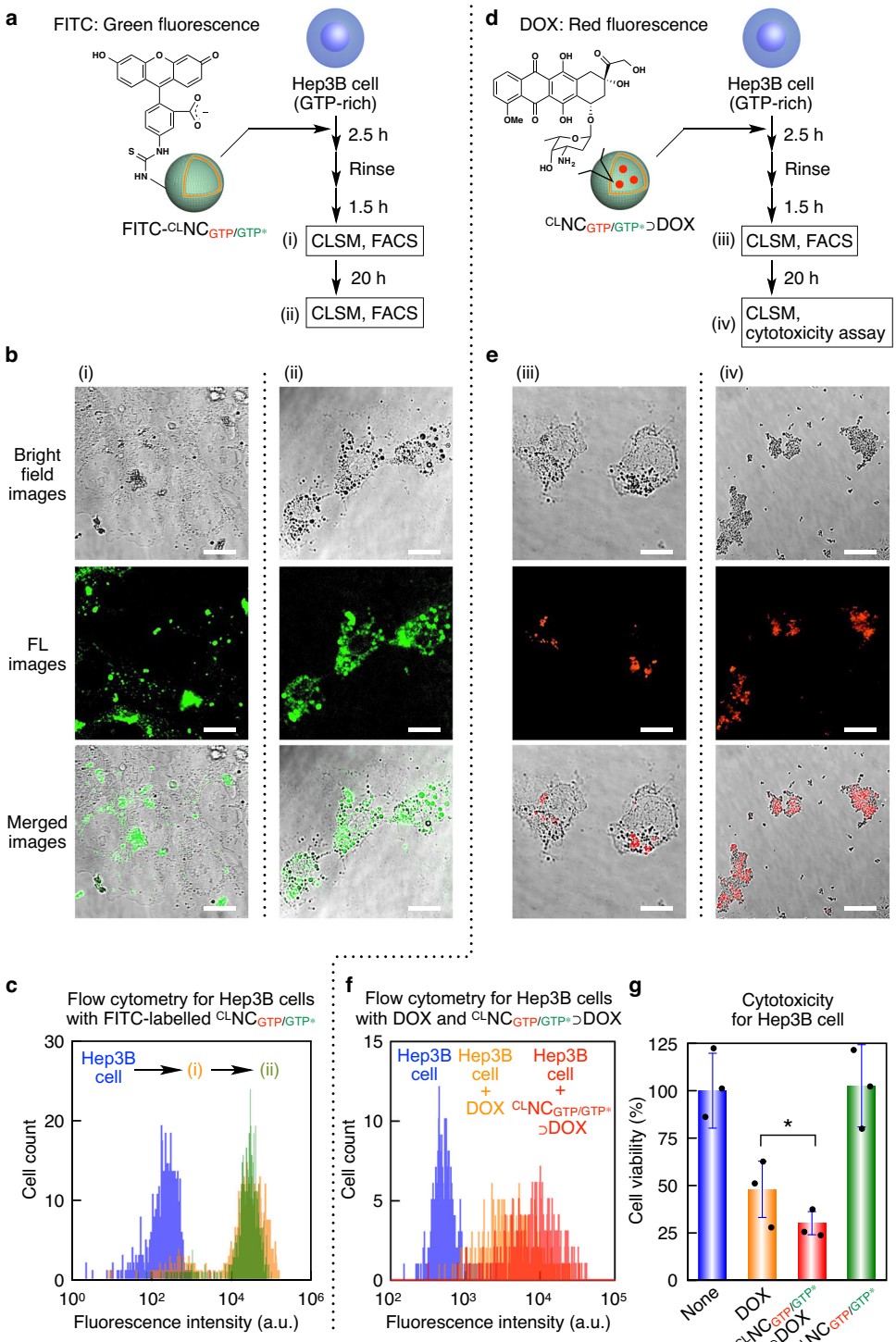

**Fig. 5 | Intracellular drug delivery using $^{CL}NC_{GTP/GTP*}$. a** Schematic illustration of the uptake of FITC-labeled $^{CL}NC_{GTP/GTP*}$ into Hep3B cells. **b** Bright field (upper row) and CLSM images displaying FITC (middle row, green) in Hep3B cells and their merged images (lower row). The cells were incubated in EMEM containing $^{CL}NC_{GTP/GTP*}$ (0.5 µg ml⁻¹) for 2.5 h, rinsed with D-PBS, and further incubated in EMEM (10% FBS) for 1.5 h (i) and 21.5 h (ii). Scale bars, 20 µm. **c** Flow cytometry profiles showing FITC fluorescence of Hep3B cells (n > 660) incubated without (blue) and with FITC-labeled $^{CL}NC_{GTP/GTP*}$ for 2.5 h, rinsed with D-PBS, and further incubated in EMEM (10% FBS) for 1.5 h (i, orange) and 21.5 h (ii, green). **d** Schematic illustration of the cellular uptake of $^{CL}NC_{GTP/GTP*}⊃DOX$. **e** Bright field (upper row) and CLSM images displaying DOX (middle row, red) in Hep3B cells

and their merged images (lower row). The cells were incubated in EMEM containing $^{CL}NC_{GTP/GTP*}⊃DOX$ ([$^{CL}NC_{GTP/GTP*}$] = 2.6 µg ml⁻¹, [DOX] = 2 µM) for 2.5 h, rinsed with D-PBS, and further incubated in EMEM (10% FBS) for 1.5 h (iii) and 21.5 h (iv). Scale bars, 20 µm. **f, g** Flow cytometry profiles (**f**) showing DOX fluorescence of Hep3B cells (n > 390) and their normalized viabilities (**g**) determined using Cell Counting Kit-8 (n = 3). The cells were incubated without (blue) and with DOX (2 µM; orange), and $^{CL}NC_{GTP/GTP*}⊃DOX$ ([$^{CL}NC_{GTP/GTP*}$] = 2.6 µg ml⁻¹, [DOX] = 2 µM; red) for 2.5 h in EMEM, and then rinsed with D-PBS, followed by incubation in EMEM (10% FBS) for 21.5 h. Statistical significance was examined by two-sided Student's t test (*p = 0.0094 < 0.01). Bars represent mean values ± SD from three different samples.

(100 mM PIPES, 2 mM $MgSO_4$, 0.5 mM GTP, 4 µg $ml^{-1}$ leupeptin, and 0.4 mM PefaBlock, pH 6.8). A solution of $THD_{GTP}$ (5.8 mg $ml^{-1}$) in PIPES buffer (100 mM PIPES, 5 mM $MgCl_2$, 2 mM $MgSO_4$, 1.5 mM GTP, and 10% DMSO, pH 6.8) was incubated at 37 °C for 30 min to afford $MT_{GTP}$. The reaction mixture was centrifuged at 17,900 × $g$ for 20 min at 24 °C. The resulting precipitate was dissolved in PIPES buffer (100 mM PIPES, 100 µM $MgCl_2$, and 20 µM GDP, pH 6.8) and incubated at 4 °C for 3 h to afford $THD_{GDP}$[43]. Subsequently, $THD_{GDP}$ (0.3 mg $ml^{-1}$) thus obtained was incubated in PIPES buffer (100 mM PIPES, 1 mM $MgCl_2$, 250 µM GTP*, and 50 µM GTP, pH 6.8) at 4 °C for 60 min and then at 37 °C for 30 min to afford $NS_{GTP/GTP*}$. $NS_{GTP/GTP*}$ (13 µg $ml^{-1}$) was incubated in a solution of $Glue^{CO_2^-}$ (100 µM) in PIPES buffer (14 mM PIPES, 1 mM $MgCl_2$, and 200 µM GTP*, pH 6.8) at 37 °C for 30 min. The reaction mixture was exposed to UV light at 300 nm for 2 min, affording $^{CL}NC_{GTP/GTP*}$. FITC-labeled $NS_{GTP/GTP*}$ and $^{CL}NC_{GTP/GTP*}$ were prepared using FITC-labeled $THD_{GDP}$ (14% labeling rate)[51] under conditions that were otherwise identical to those listed above. Prior to the NMR measurement of the $NS_{GTP/GTP*}$ sample, unbound GTP and GTP* were removed by centrifugation (286,000 × $g$) of the reaction mixture at 37 °C for 60 min. Zeta potentials of $NS_{GTP/GTP*}$ (1.3 µg $ml^{-1}$) and $NC_{GTP/GTP*}$ (1.3 µg $ml^{-1}$) were measured at 37 °C in PIPES buffer.

### GTP-responsiveness of $^{CL}NC_{GTP/GTP*}$

A solution of $^{CL}NC_{GTP/GTP*}$ (12 µg $ml^{-1}$) in PIPES buffer (9 mM PIPES, 0.9 mM $MgCl_2$, and 180 µM GTP*, pH 6.8) was incubated in the presence of GTP (0.1 mM, 0.2 mM, 0.5 mM, and 1 mM), ATP (0.5 mM), CTP (0.5 mM), and UTP (0.5 mM) at 37 °C for 100 min. For the evaluation of the GTP hydrolysis activities of $^{CL}NC_{GTP/GTP*}$ (12 µg $ml^{-1}$) and $THD_{GDP}$ (12 µg $ml^{-1}$), Biomol Green™ reagent (100 µl) was added to the reaction mixtures, incubated for 30 min at room temperature and subjected to electronic absorption spectroscopy at 620 nm.

### Preparation of $^{CL}NC_{GTP/GTP*}{\supset}NP_{Au}$

$^{CL}NC_{GTP/GTP*}{\supset}NP_{Au}$ was prepared after the incubation of a mixture of $NS_{GTP/GTP*}$ (13 µg $ml^{-1}$), $Glue^{CO_2^-}$ (100 µM), and gold nanoparticles ($NP_{Au}$; 14 pM) in PIPES buffer (14 mM PIPES, 1 mM $MgCl_2$, and 200 µM GTP*, pH 6.8) at 37 °C for 30 min, followed by UV irradiation at 300 nm for 2 min. For the asymmetric field flow fractionation analysis, a sample solution of $^{CL}NC_{GTP/GTP*}{\supset}NP_{Au}$ in PIPES buffer was subjected to ultrafiltration (1500 × $g$) for 5 min using a regenerated cellulose membrane (cut-off MW = 5000) prior to analysis. PEG-coated $NP_{Au}$ was used to avoid nonspecific adhesion of THD[52].

### GTP-triggered release of DOX from $^{CL}NC_{GTP/GTP*}$

$^{CL}NC_{GTP/GTP*}{\supset}DOX$ was prepared after the incubation of a mixture of $NS_{GTP/GTP*}$ (13 µg $ml^{-1}$), $Glue^{CO_2^-}$ (100 µM), and DOX (10 µM) in PIPES buffer (14 mM PIPES, 1 mM $MgCl_2$, and 400 µM GTP*, pH 6.8) at 37 °C for 30 min, followed by UV irradiation at 300 nm for 2 min. The reaction mixture was incubated with GTP (1 mM) at 37 °C for 100 min and then subjected to ultrafiltration (2400 × $g$) using a regenerated cellulose membrane (cut-off MW = 5000) for 10 min. The resulting residue was subjected to fluorescence spectroscopy ($\lambda_{ext}$ = 470 nm). A reference sample without GTP was likewise prepared.

### Intracellular delivery

Hep3B cells (3.0 × $10^3$ cells/well) plated onto an 8-well chambered cover glass were incubated in EMEM containing 10% FBS at 37 °C with 5% $CO_2$ for 24 h. The cell samples were rinsed twice with D-PBS prior to use. Typically, the cells were treated with FITC-labeled $^{CL}NC_{GTP/GTP*}$ (0.5 µg $ml^{-1}$) and incubated at 37 °C with 5% $CO_2$ for 2.5 h. Then, the cells were rinsed twice with D-PBS and further incubated at 37 °C for 1.5 h (4-h incubation in total) or 21.5 h (24-h incubation in total) with 5% $CO_2$ in EMEM containing 10% FBS. Analogous cell samples treated with FITC-labeled $THD_{GDP}$ (0.5 µg $ml^{-1}$), FITC-labeled $NS_{GTP/GTP*}$ (0.5 µg $ml^{-1}$), $^{CL}NC_{GTP/GTP*}$ (0.5 µg $ml^{-1}$) with $NaN_3$ (5 mM)[49], $^{CL}NC_{GTP/GTP*}{\supset}DOX$

([$^{CL}NC_{GTP/GTP*}$] = 2.6 µg $ml^{-1}$, [DOX] = 2 µM), $^{CL}NC_{GTP/GTP*}$ (2.6 µg $ml^{-1}$), and DOX (2 µM) were likewise prepared. For a cell viability assay using $^{CL}NC_{GTP/GTP*}{\supset}DOX$, $^{CL}NC_{GTP/GTP*}$, and DOX, the cell samples were incubated with Cell Counting Kit-8 reagents (10 µl) for 30 min, and subjected to electronic absorption spectroscopy at 450 nm. Hep3B cell samples treated with Tween 20 (0.2%) were used as a positive control.

### Statistics and reproducibility

All experiments including the preparation of $^{CL}NC_{GTP/GTP*}$, the investigation of its GTP-responsive collapse, and the intracellular delivery using $^{CL}NC_{GTP/GTP*}$ were performed at least three times to check the reproducibility.

### Reporting summary

Further information on research design is available in the Nature Research Reporting Summary linked to this article.

## Data availability

All the data corresponding to the findings of this study are provided in the article and Supplementary Information. Source data is available for Figs. 2b, 3g, i, j, 4c–e, h, 5c, f, and g and Supplementary Figs. 9–11, 13–16, 19–25, 27, 28 and 30–36 in the associated source data file. 3D structures of THD for the MD simulation were obtained from Protein Data Bank (PDB) (PDB code: 3J6E and 1TUB). Complete modeling data, structures and parameters used for, and extracted from simulations are available at https://zenodo.org/record/7070651#.Yx80t9JBxkg. Source data are provided with this paper.

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

## Acknowledgements

This work was financially supported by a JSPS Grant-in-Aid for Scientific Research (S) (18H05260 to T.A.) and JSPS KAKENHI Early-Career Scientists (19K15378 to N.U.). We also acknowledge Japan Association for Chemical Innovation and Moritani foundation for N.U. A.K. thanks the WINGS/GPLLI Collaboration Project (the Univ. of Tokyo) and Research Fellowships of JSPS for Young Scientists. K.O. acknowledges the support by the CAS-Croucher Funding Scheme for Joint Laboratories. G.M.P. acknowledges the support received by the European Research Council (ERC) under the European Union's Horizon 2020 research and innovation program (grant agreement no. 818776 – DYNAPOL). The small-angle X-ray scattering measurements were performed at BL45XU in SPring-8 with the approval of the RIKEN SPring-8 Center. The authors also acknowledge the computational resources provided by the Swiss National Supercomputing Center (CSCS) and by CINECA.

## Author contributions

N.U., A.K., and K.O. designed and performed all experiments; A.C., C.L., E.A.Z., M.D., and G.M.P. performed and analyzed the MD simulation; N.U. and M.T. prepared THD; T.H. supported the small-angle X-ray scattering measurements at SPring-8; N.U., G.M.P., and T.A. analyzed the data and wrote the manuscript.

## Competing interests

The authors declare no competing interests.
