## [Peer Review File · Nature Communications]

Reconstitution of microtubule into GTP-responsive nanocapsulesREVIEWER COMMENTS

Reviewer #1 (Remarks to the Author):

The article by Uchida et al reports the synthesis, characterization and application of GTP-responsive nanocapsules. In summary, they find that assembly of two THD monomers that differ in the nature of the GTP moiety results in the formation of a 2D nanosheet. This 2D nanosheet can be stabilized by molecular glue and used in biological media without undergoing an (osmotic?)-induced collapse. In the last part of the paper, the authors employ this technology for drug delivery in cancer cell lines. Overall, the publication is of high quality. The novelty of the research lies in the development of a novel GTP-responsive nanostructure that does not respond to other triphosphates. To improve the article, I suggest the following additions:

- 1) The authors should discuss the reason why only for certain molar ratios of THD-GTP and THD-GTP* a 2D nanosheet is formed. In the current state, there is only a single speculative sentence, but I think a longer discussion perhaps supported by MD simulations should be more appropriate. It is one of the key findings of the work and needs to be better addressed. By speculating the underlying mechanism, this remarkable transformation could be more generalized.
- 2) All figures containing bar graphs should be replaced with graphs showing the distribution of the data points (for example a beeswarm plot). In addition, the number of replicates should be added to the caption of each Figure caption. For cell experiments, an additional statement should be made about whether the replicates are biological or technical.
- 3) Can the authors speculate on how changes at the molecular level could be used to tune the GTP concentration where collapse occurs? For example, does the cross-link density correlate to the GTP concentration where the capsules disintegrate?

Reviewer #2 (Remarks to the Author):

In this manuscript, Uchida et al. developed the nanocapsules by reconstitution of microtubule with the molecular glue. They demonstrated the nanocapsules could respond to GTP and thus release the encapsulated DOX. The overall study is well organized and performed. Thus I enthusiastically support acceptance of this work after addressing following issues.

1. Although the authors demonstrated the nanocapsules could release DOX after a 100-min incubation with GTP, how about its detailed release profile? Please incorporate.
2. How about the stability of the nanocapsules under different pH, especially in an acidic condition within the tumor tissue?
3. The cell cytotoxicity of DOX-loaded nanocapsules should be further carefully investigated, such as comparison of different concentrations and different incubation time, even in different cancer cell lines.
4. The authors stated that these GTP-responsive nanocapsules could selectively kill cancer cells. They should investigate the cytotoxicity of DOX-loaded nanocapsules towards normal cells.
5. The in vivo performance of this system is preferred to studied.

Reviewer #3 (Remarks to the Author):

This manuscript describes the preparation of GTP-responsive nanocapsules from microtubule.

Tubulin nanosheet was formed by incubating tubulin monomer in the presence of a mixture of GTP (17 mol%) and nonhydrolysable GTP* (83 mol%). The addition of a photoreactive molecular glue transformed the tubulin nanosheet complexed with GTP/GTP* to a GTP-responsive nanocapsule. The authors encapsulated doxorubicin into this nanostructure and delivered it into cancer cells. The work is novel and is expected to be of interest to a broad range of researchers. Most of the claims are backed by the experimental results and the paper is well written. However, there are some points that should be addressed before its publication.

1) On page 3, the author claimed that GTP* can be replaced by GTP: "In contrast, GTP attached to the β -tubulin unit is known to be hydrolysable and even replaceable with, e.g., GTP*, a nonhydrolysable GTP analogue (guanylyl 5'- α , β -methylenediphosphonate), affording THDGTP* (for convenience, only variable nucleoside phosphates attached to the β -tubulin unit are shown as a subscript)." It is not clear whether this has been proven in the past or if the authors are referring to their experience with the system. In any case they should clarify this point and if it is something they have observed, it should be clearly supported by experimental results.

2) The statement on page 5 line 12 might be misleading, "By means of nuclear magnetic resonance (NMR) spectroscopy in DMSO, NSGTP/GTP* prepared at a GTP* content of 83 mol% was found to contain 65 mol% of THDGTP* (Supplementary Fig. 12), suggesting that THDGTP assembles more preferentially than THDGTP* into NS GTP/GTP*". The lower GTP* content could also result from low binding affinity of GTP* to THD. Overall, the quantification by NMR spectroscopy is tricky, because it is performed in a different solvent and there is the question whether the signal is coming from free or weakly bound GTP*.

3) How was GTP hydrolysis prevented during the preparation of the nanocapsule?

4) The decomposition rate of the nanocapsules in the presence of varying concentrations of GTP is an important factor for drug delivery thus needs to be evaluated.

5) To demonstrate the selectivity of GTP over ATP, a negative control such as drug release in low-GTP-expressing cells might be necessary.

6) It would be important to know whether the intracellular delivery of DOX is induced by the nanocapsule itself or from its fragments. Therefore, control cytotoxicity experiments should be performed with non-assembled THD(GDP) and THD(GDP)+Glue(CO₂-). This would help clarify the role of the assembled structure plays in the delivery.

7) Although the molecular dynamics simulations are very interesting and insightful, the author might want to make a stronger connection between the change of curvature induced by the molecular glue on a nanosheet and the final spherical morphology of the assembly. This point isn't really clear in the current version of the manuscript.

Point-by-Point Response to Reviewer's Comments

For Reviewer 1

[1] Overall, the publication is of high quality. The novelty of the research lies in the development of a novel GTP-responsive nanostructure that does not respond to other triphosphates.

=> We appreciate the reviewer's highly encouraging remark.

[2] The authors should discuss the reason why only for certain molar ratios of THD_{GTP} and THD_{GTP*} a 2D nanosheet is formed. In the current state, there is only a single speculative sentence, but I think a longer discussion perhaps supported by MD simulations should be more appropriate. It is one of the key findings of the work and needs to be better addressed. By speculating the underlying mechanism, this remarkable transformation could be more generalized.

=> We appreciate the reviewer's constructive suggestion. It is known that MT_{GTP} and MT_{GTP*}, prepared from THD_{GTP} and THD_{GTP*}, respectively, are formed by edge-closing of NS_{GTP} and NS_{GTP*} as transient precursors (*Cell Cycle* **2005**, *4*, 1157–1160). Note that the longer axis of THD_{GTP} is shorter than that of THD_{GTP*}. We suppose that this mismatch possibly affords unfolded NS_{GTP/GTP*} rather than folded MT_{GTP/GTP*}. Indeed, when THD_{GTP*} (GTP* content of 83 mol%) was coassembled with THD_{GTP γ S} (THD hybridized with guanosine 5'-O-(3-thiotriphosphate), GTP γ S), whose length is likewise shorter than THD_{GTP*}, NS_{GTP γ S/GTP*} was formed (Supplementary Fig. 13), whereas the coassembly of THD_{GTP} and THD_{GTP γ S} (GTP γ S content of 83 mol%), whose longer axes are close in length to each other, resulted in MT_{GTP/GTP γ S} (Supplementary Fig. 14). In the revised manuscript, we discussed the mechanism based on the above results.

[3] All figures containing bar graphs should be replaced with graphs showing the distribution of the data points (for example a beeswarm plot). In addition, the number of replicates should be added to the caption of each Figure caption. For cell experiments, an additional statement should be made about whether the replicates are biological or technical.

=> We revised the figures and captions, and added the following sentence: "Bars represent mean values \pm SD from three different samples." In regard to Fig. 3g, we did not add data points because their number is too large (> 1,000).

[4] Can the authors speculate on how changes at the molecular level could be used to tune the GTP concentration where collapse occurs? For example, does the cross-link density correlate to the GTP concentration where the capsules disintegrate?

=> We prepared a photo-crosslinked version of ^{CL}NC_{GTP/GTP*}, which clearly showed an enhanced durability toward albumin and serum under physiological conditions, but collapsed, just like non-

crosslinked $\text{NC}_{\text{GTP/GTP}^*}$, upon treatment with GTP (Fig. R1). We therefore consider that the crosslinked part with $\text{Glue}^{\text{CO}_2^-}$ is different from the GTP-responsive part.

Fig. R1. TEM image of $\text{NC}_{\text{GTP/GTP}^*}$ ($9 \mu\text{g ml}^{-1}$) after a 100-min incubation with 0.5 mM of GTP.

For Reviewer 2

[1] The overall study is well organized and performed. Thus, I enthusiastically support acceptance of this work after addressing following issues.

=> We appreciate these highly encouraging remarks.

[2] Although the authors demonstrated the nanocapsules could release DOX after a 100-min incubation with GTP, how about its detailed release profile? Please incorporate.

=> We newly added the release profiles of DOX upon 20-min and 50-min incubations to that upon 100-min incubation with 1 mM GTP (Fig 4h). The results allow us to confirm that the released amount of DOX from ${}^{\text{CL}}\text{NC}_{\text{GTP/GTP}^*}$ increased as the incubation time with GTP was increased. We added this description to the revised manuscript.

[3] How about the stability of the nanocapsules under different pH, especially in an acidic condition within the tumor tissue?

=> $\text{THD}_{\text{GTP/GTP}^*}$ is known to change its conformation at $\text{pH} < 6.2$, as a consequence of the detachment of GTP and GTP^* from the constituent β -tubulin unit (*Biochemistry* **1992**, *31*, 10610). However, as shown in Fig. 2h, ${}^{\text{CL}}\text{NC}_{\text{GTP/GTP}^*}$ remained unchanged in PIPES buffer at pH 6.8, which is in a range of pH at tumor tissue ($\text{pH} = 6.5\text{--}7.0$; *Cancer Res.* **1996**, *56*, 1194). When the pH of the medium was lowered to pH 5.5, ${}^{\text{CL}}\text{NC}_{\text{GTP/GTP}^*}$, as observed by TEM, decomposed upon 60-min incubation (Fig. R2).

Fig. R2. TEM image of ${}^{\text{CL}}\text{NC}_{\text{GTP/GTP}^*}$ ($13 \mu\text{g ml}^{-1}$) upon incubation in PIPES buffer at pH 5.5 for 60 min.

[4] The cell cytotoxicity of DOX-loaded nanocapsules should be further carefully investigated, such as comparison of different concentrations and different incubation time, even in different cancer cell lines.

=> We newly conducted the cytotoxicity tests of ${}^{\text{CL}}\text{NC}_{\text{GTP}/\text{GTP}^*} \supset \text{DOX}$ at various concentrations (Supplementary Fig. 33) and incubation times (Supplementary Fig. 34) using two additional cancer cell lines (Supplementary Fig. 35). Accordingly, we added the following sentences to the revised manuscript: "As expected, the cell viability decreased as the concentration of ${}^{\text{CL}}\text{NC}_{\text{GTP}/\text{GTP}^*} \supset \text{DOX}$ was increased (Supplementary Fig. 33), while the viability upon pre-treatment with ${}^{\text{CL}}\text{NC}_{\text{GTP}/\text{GTP}^*} \supset \text{DOX}$ for 2.5 h did not substantially increase when the pre-treatment time was shortened to 1.0 h (Supplementary Fig. 34). The intracellular delivery of ${}^{\text{CL}}\text{NC}_{\text{GTP}/\text{GTP}^*} \supset \text{DOX}$ was also successful with other cell lines such as A549 cell and HeLa cell (Supplementary Fig. 35)."

[5] The authors stated that these GTP-responsive nanocapsules could selectively kill cancer cells. They should investigate the cytotoxicity of DOX-loaded nanocapsules towards normal cells.

=> We do not state that "these GTP-responsive nanocapsules could selectively kill cancers."

=> Since normal cells are known to be intact to the treatment with DOX (*Science* **2011**, 334, 1129), they are most likely intact to the treatment with DOX-loaded nanocapsules, considering also the result that the nanocapsules themselves are nontoxic.

[6] The *in vivo* performance of this system is preferred to be studied.

=> Although we are interested in the *in vivo* performance, we would very much appreciate it if this reviewer and handling editor could allow us to leave this study to our future investigation, because it may take more than a year to accomplish.

For Reviewer 3

[1] The work is novel and is expected to be of interest to a broad range of researchers. Most of the claims are backed by the experimental results and the paper is well written.

=> We appreciate the reviewer's highly encouraging remarks.

[2] On page 3, the author claimed that GTP* can be replaced by GTP: "In contrast, GTP attached to the β -tubulin unit is known to be hydrolysable and even replaceable with, *e.g.*, GTP*, a nonhydrolysable GTP analogue (guanylyl 5'- α,β -methylenediphosphonate), affording THD_{GTP*} (for convenience, only variable nucleoside phosphates attached to the β -tubulin unit are shown as a subscript)." It is not clear whether this has been proven in the past or if the authors are referring to their experience with the system. In any case they should clarify this point and if it is something they have observed, it should be clearly supported by experimental results.

=> The phenomenon pointed out is our new finding. Although THD_{GTP*} has no GTPase activity, the product upon incubation of THD_{GTP*} with GTP for 1 h in PIPES at 37 °C showed a GTPase activity comparable to that of THD_{GTP} (Supplementary Fig. 25), indicating the conversion of THD_{GTP*} into THD_{GTP}. In the revised manuscript, we clarified this point.

[3] The statement on page 5 line 12 might be misleading, "By means of nuclear magnetic resonance (NMR) spectroscopy in DMSO, NS_{GTP/GTP*} prepared at a GTP* content of 83 mol% was found to contain 65 mol% of THD_{GTP*} (Supplementary Fig. 12), suggesting that THD_{GTP} assembles more preferentially than THD_{GTP*} into NS_{GTP/GTP*}". The lower GTP* content could also result from low binding affinity of GTP* to THD. Overall, the quantification by NMR spectroscopy is tricky, because it is performed in a different solvent and there is the question whether the signal is coming from free or weakly bound GTP*.

=> We appreciate the reviewer's comment. To avoid misleading, we removed the following description: "suggesting that THD_{GTP} assembles more preferentially than THD_{GTP*} into NS_{GTP/GTP*}."

=> As described in the Methods section, the sample employed for the ¹H NMR spectroscopy had been centrifuged for the removal of free GTP*. So, a sharp singlet signal in DMSO-*d*₆, which was identified to be OPCH₂PO using the authentic sample of GTP* (marked as a blue circle in Supplementary Fig. 12), must originate from GTP* attached to THD, although it is uncertain whether this GTP* is free or weakly bound in DMSO-*d*₆.

[4] How was GTP hydrolysis prevented during the preparation of the nanocapsule?

=> Preparation of ¹⁴C_{GTP/GTP*} takes roughly 1 hour. We do not know how much the GTP hydrolysis, if any occurs during the preparation, affects the stability of ¹⁴C_{GTP/GTP*}. Nevertheless, we confirmed that ¹⁴C_{GTP/GTP*} remained unchanged without any deterioration for

3 hours (Supplementary Fig. 27) when it was unexposed to GTP. This is certainly because non-hydrolyzable GTP* is dominant (65%) in ${}^{\text{CL}}\text{NC}_{\text{GTP}/\text{GTP}^*}$.

[5] The decomposition rate of the nanocapsules in the presence of varying concentrations of GTP is an important factor for drug delivery thus needs to be evaluated.

=> Please note that our original Fig. 4c shows the DLS profiles of ${}^{\text{CL}}\text{NC}_{\text{GTP}/\text{GTP}^*}$ upon incubation in PIPES buffer with varying concentrations of GTP for 100 min (0, 0.2, 0.5, and 1.0 mM) at 37 °C.

=> We newly added to the revised manuscript the release profile of DOX from ${}^{\text{CL}}\text{NC}_{\text{GTP}/\text{GTP}^*} \rightarrow \text{DOX}$ upon incubation at 37 °C in PIPES buffer containing 1 mM GTP, where the %-release value of DOX increased from 27% to 47% and 79% in 20 min, 50 min, and 100 min, respectively (Fig. 4h).

[6] To demonstrate the selectivity of GTP over ATP, a negative control such as drug release in low-GTP-expressing cells might be necessary.

=> Since low-GTP-expressing cells such as normal cells are known to be intact to the treatment with DOX (*Science* **2011**, 334, 1129), they are most likely intact to the treatment with DOX-loaded nanocapsules, considering also the result that the nanocapsules themselves are nontoxic.

[7] It would be important to know whether the intracellular delivery of DOX is induced by the nanocapsule itself or from its fragments. Therefore, control cytotoxicity experiments should be performed with non-assembled THD_{GDP} and $\text{THD}_{\text{GDP}} + \text{Glue}^{\text{CO}_2^-}$. This would help clarify the role of the assembled structure plays in the delivery.

=> We confirmed that neither the coexistence of THD_{GDP} nor $\text{THD}_{\text{GDP}}/\text{Glue}^{\text{CO}_2^-}$ enhanced the efficacy of DOX (Supplementary Fig. 36).

[8] Although the molecular dynamics simulations are very interesting and insightful, the author might want to make a stronger connection between the change of curvature induced by the molecular glue on a nanosheet and the final spherical morphology of the assembly. This point isn't really clear in the current version of the manuscript.

=> We appreciate the reviewer's constructive suggestion. In response to the reviewer's suggestion, we added the following sentences to the revised manuscript: The computational calculation showed that multiple adhering $\text{Glue}^{\text{CO}_2^-}$ molecules can interact and self-assemble via a salt-bridge interaction between their Gu^+ and CO_2^- groups (Fig. 3j, red), which results in forming a dense $\text{Gu}^+/\text{CO}_2^-$ salt-bridged polymeric network on the $[\text{THD}_{\text{GTP}^*}]_3$ surface (Fig. 3k). This may promote the self-assembly of flexible $\text{NS}_{\text{GTP}/\text{GTP}}$ and stabilize them in the gently curved multilayered configuration of $\text{NC}_{\text{GTP}/\text{GTP}^*}$, as observed experimentally^{46,47}.

REVIEWERS' COMMENTS

Reviewer #1 (Remarks to the Author):

The authors have answered all my points in a consistent manner. The article can be accepted as it is.

Reviewer #2 (Remarks to the Author):

The authors have addressed my comments.

Reviewer #3 (Remarks to the Author):

In the revised version of the manuscript "Reconstitution of microtubule into GTP-responsive nanocapsules" the authors have addressed most of the points raised by the reviewers. I particularly appreciate the addition of control experiments that elucidate the effects and the therapeutic potential of the new nanostructure on cells (Supplementary Figure 33, 34, 35, and 36).

The manuscript is at this point suitable for publication in Nature Communications and I am confident it will be of interest to the community.

There are a couple of points that the authors have not been able to address and I would like to point them out so they can consider them.

1. The reviewer question: To demonstrate the selectivity of GTP over ATP, a negative control such as drug release in low-GTP-expressing cells might be necessary. =>

The authors reply: Since low-GTP-expressing cells such as normal cells are known to be intact to the treatment with DOX (Science 2011, 334, 1129), they are most likely intact to the treatment with DOX-loaded nanocapsules, considering also the result that the nanocapsules themselves are nontoxic.

My new comment: There is probably a misunderstanding on the original question. It is not clear in the manuscript how the speculation presented by the author addresses the point of selectivity of the system of GTP over ATP.

2. Reviewer comment: The statement on page 5 line 12 might be misleading, "By means of nuclear magnetic resonance (NMR) spectroscopy in DMSO, NSGTP/GTP* prepared at a GTP* content of 83 mol% was found to contain 65 mol% of THDGTP* (Supplementary Fig. 12), suggesting that THDGTP* assembles more preferentially than THDGTP* into NSGTP/GTP*". The lower GTP* content could also result from low binding affinity of GTP* to THD. Overall, the quantification by NMR spectroscopy is tricky, because it is performed in a different solvent and there is the question whether the signal is coming from free or weakly bound GTP*.

Author reply: => We appreciate the reviewer's comment. To avoid misleading, we removed the following description: "suggesting that THDGTP* assembles more preferentially than THDGTP* into NSGTP/GTP*."

As described in the Methods section, the sample employed for the ¹H NMR spectroscopy had been centrifuged for the removal of free GTP*. So, a sharp singlet signal in DMSO-d₆, which was identified to be OPCH₂PO using the authentic sample of GTP* (marked as a blue circle in Supplementary Fig. 12), must originate from GTP* attached to THD, although it is uncertain whether this GTP* is free or weakly bound in DMSO-d₆

My new comment: I appreciate that the authors acknowledge the limitation of NMR in giving a molecular picture of the system, however the authors contradict themselves saying that the signal must originate from GTP* attached to THD but at the same time acknowledging that they don't know whether this GTP* is free or bound. As a matter of fact, there is no insight on whether this GTP* is

bound or not and the centrifugation without a specified speed does not guarantee there is no more bound GTP*, it is only going to decrease the total concentration of GTP* changing the equilibrium possibly towards the unbound state.

Point-by-Point Response to Reviewer #3's Comments

1. The reviewer question: To demonstrate the selectivity of GTP over ATP, a negative control such as drug release in low-GTP-expressing cells might be necessary. =>

The authors reply: Since low-GTP-expressing cells such as normal cells are known to be intact to the treatment with DOX (*Science* **2011**, 334, 1129), they are most likely intact to the treatment with DOX-loaded nanocapsules, considering also the result that the nanocapsules themselves are nontoxic.

(1) There is probably a misunderstanding on the original question. It is not clear in the manuscript how the speculation presented by the author addresses the point of selectivity of the system of GTP over ATP.

=> Considering the suggestion by this reviewer, we toned down our statement by partly revising the original manuscript on page 11 (see below in the **Revised Part**, where red-colored sentences have been revised). Although the particular GTP/ATP selectivity observed for $^{13}\text{C}_{\text{GTP/GTP}^*}$ was from a test-tube experiment at present, we have a strong belief that this achievement, even without cell-based experiments (requiring long time), deserves publication as it is highly important but has never been achieved.

Revised Part: This is likely caused by the GTP-selective collapse of $^{13}\text{C}_{\text{GTP/GTP}^*}$. The intracellular delivery of $^{13}\text{C}_{\text{GTP/GTP}^*} \rightarrow \text{DOX}$ was also successful with other cell lines such as A549 cell and HeLa cell (Supplementary Fig. 35). We also confirmed that neither the coexistence of THD_{GDP} nor $\text{THD}_{\text{GDP}}/\text{Gluc}^{\text{CO}_2^-}$ enhanced the efficacy of DOX (Supplementary Fig. 36). Together with the noncytotoxic nature of $^{13}\text{C}_{\text{GTP/GTP}^*}$ (Fig. 5g, green) and its stability in a range of pH at tumor tissue (Fig. 2h)⁵⁰, these results allow us to expect that $^{13}\text{C}_{\text{GTP/GTP}^*}$ may have the potential to deliver preloaded drugs into cancer cells using GTP as an endogenous reporter.

2. Reviewer comment: The statement on page 5 line 12 might be misleading, "By means of nuclear magnetic resonance (NMR) spectroscopy in DMSO, $\text{NS}_{\text{GTP/GTP}^*}$ prepared at a GTP* content of 83 mol% was found to contain 65 mol% of $\text{THD}_{\text{GTP}^*}$ (Supplementary Fig. 12), suggesting that THD_{GTP} assembles more preferentially than $\text{THD}_{\text{GTP}^*}$ into $\text{NS}_{\text{GTP/GTP}^*}$ ". The lower GTP* content could also result from low binding affinity of GTP* to THD. Overall, the quantification by NMR spectroscopy is tricky, because it is performed in a different solvent and there is the question whether the signal is coming from free or weakly bound GTP*.

Author reply: => We appreciate the reviewer's comment. To avoid misleading, we removed the following description: "suggesting that THD_{GTP} assembles more preferentially than THD_{GTP*} into NS_{GTP/GTP*}."

As described in the Methods section, the sample employed for the ¹H NMR spectroscopy had been centrifuged for the removal of free GTP*. So, a sharp singlet signal in DMSO-*d*₆, which was identified to be OPCH₂PO using the authentic sample of GTP* (marked as a blue circle in Supplementary Fig. 12), must originate from GTP* attached to THD, although it is uncertain whether this GTP* is free or weakly bound in DMSO- *d*₆

(2) I appreciate that the authors acknowledge the limitation of NMR in giving a molecular picture of the system, however the authors contradict themselves saying that the signal must originate from GTP* attached to THD but at the same time acknowledging that they don't know whether this GTP* is free or bound. As a matter of fact, there is no insight on whether this GTP* is bound or not and the centrifugation without a specified speed does not guarantee there is no more bound GTP*, it is only going to decrease the total concentration of GTP* changing the equilibrium possibly towards the unbound state.

=> It was reported that the binding between GTP* and tubulin is kinetically stable in buffer solutions and GTP* hardly dissociates from tubulin during centrifugation (*J. Cell. Biol.* **2012**, *198*, 315). We followed this paper to design our experiments.